# Bayesian Dyadic Trees and Histograms for Regression

**Stéphanie van der Pas**
Mathematical Institute
Leiden University
Leiden, The Netherlands
svdpas@math.leidenuniv.nl

**Veronika Ročková**
Booth School of Business
University of Chicago
Chicago, IL, 60637
Veronika.Rockova@ChicagoBooth.edu

## Abstract

Many machine learning tools for regression are based on recursive partitioning of the covariate space into smaller regions, where the regression function can be estimated locally. Among these, regression trees and their ensembles have demonstrated impressive empirical performance. In this work, we shed light on the machinery behind Bayesian variants of these methods. In particular, we study Bayesian regression histograms, such as Bayesian dyadic trees, in the simple regression case with just one predictor. We focus on the reconstruction of regression surfaces that are piecewise constant, where the number of jumps is unknown. We show that with suitably designed priors, posterior distributions concentrate around the true step regression function at a near-minimax rate. These results *do not* require the knowledge of the true number of steps, nor the width of the true partitioning cells. Thus, Bayesian dyadic regression trees are fully adaptive and can recover the true piecewise regression function nearly as well as if we knew the exact number and location of jumps. Our results constitute the first step towards understanding why Bayesian trees and their ensembles have worked so well in practice. As an aside, we discuss prior distributions on balanced interval partitions and how they relate to an old problem in geometric probability. Namely, we relate the probability of covering the circumference of a circle with random arcs whose endpoints are confined to a grid, a new variant of the original problem.

## 1 Introduction

Histogram regression methods, such as regression trees [1] and their ensembles [2], have an impressive record of empirical success in many areas of application [3, 4, 5, 6, 7]. Tree-based machine learning (ML) methods build a piecewise constant reconstruction of the regression surface based on ideas of recursive partitioning. Perhaps the most popular partitioning schemes are the ones based on parallel-axis splits. One recent example is the Mondrian process [8], which was introduced to the ML community as a prior over tree data structures with interesting self-consistency properties. Many efficient algorithms exist that can be deployed to fit regression histograms underpinned by some partitioning scheme. Among these, Bayesian variants, such as Bayesian CART [9, 10] and BART [11], have appealed to umpteen practitioners. There are several reasons why. Bayesian tree-based regression tools (a) can adapt to regression surfaces without any need for pruning, (b) are reluctant to overfit, (c) provide an avenue for uncertainty statements via posterior distributions. While practical success stories abound [3, 4, 5, 6, 7], the theoretical understanding of Bayesian regression tree methods has been lacking. In this work, we study the quality of posterior distributions with regard to the three properties mentioned above. We provide first theoretical results that contribute to the understanding of Bayesian Gaussian regression methods based on recursive partitioning.

Our performance metric will be the speed of posterior concentration/contraction around the true regression function. This is ultimately a frequentist assessment, describing the typical behavior of the posterior under the true generative model [12]. Posterior concentration rate results are now slowly

entering the machine learning community as a tool for obtaining more insights into Bayesian methods [13, 14, 15, 16, 17]. Such results quantify not only the typical distance between a point estimator (posterior mean/median) and the truth, but also the typical spread of the posterior around the truth. Ideally, most of the posterior mass should be concentrated in a ball centered around the true value with a radius proportional to the minimax rate [12, 18]. Being inherently a performance measure of both location and spread, optimal posterior concentration provides a necessary certificate for further uncertainty quantification [19, 20, 21]. Beyond uncertainty assessment, theoretical guarantees that describe the average posterior shrinkage behavior have also been a valuable instrument for assessing the suitability of priors. As such, these results can often provide useful guidelines for the choice of tuning parameters, e.g. the latent Dirichlet allocation model [14].

Despite the rapid growth of this frequentist-Bayesian theory field, posterior concentration results for Bayesian regression histograms/trees/forests have, so far, been unavailable. Here, we adopt this theoretical framework to get new insights into why these methods work so well.

**Related Work**

Bayesian density estimation with step functions is a relatively well-studied problem [22, 23, 24]. The literature on Bayesian histogram regression is a bit less crowded. Perhaps the closest to our conceptual framework is the work by Coram and Lalley [25], who studied Bayesian non-parametric binary regression with uniform mixture priors on step functions. The authors focused on $L_1$ consistency. Here, we focus on posterior concentration rather than consistency. We are not aware of any other related theoretical study of Bayesian histogram methods for Gaussian regression.

**Our Contributions**

In this work we focus on a canonical regression setting with merely one predictor. We study hierarchical priors on step functions and provide conditions under which the posteriors concentrate optimally around the true regression function. We consider the case when the true regression function itself is a step function, i.e. a tree or a tree ensemble, where the number and location of jumps is unknown.

We start with a very simple space of approximating step functions, supported on equally sized intervals where the number of splits is equipped with a prior. These partitions include dyadic regression trees. We show that for a suitable complexity prior, all relevant information about the true regression function (jump sizes and the number of jumps) is learned from the data automatically. During the course of the proof, we develop a notion of the complexity of a piecewise constant function relative to its approximating class.

Next, we take a larger approximating space consisting of functions supported on balanced partitions that do not necessarily have to be of equal size. These correspond to more general trees with splits at observed values. With a uniform prior over all balanced partitions, we are able to achieve a nearly ideal performance (as if we knew the number and the location of jumps). As an aside, we describe the distribution of interval lengths obtained when the splits are sampled uniformly from a grid. We relate this distribution to the probability of covering the circumference of a circle with random arcs, a problem in geometric probability that dates back to [26, 27]. Our version of this problem assumes that the splits are chosen from a discrete grid rather than from a unit interval.

**Notation**

With $\propto$ and $\lesssim$ we will denote an equality and inequality, up to a constant. The $\varepsilon$-covering number of a set $\Omega$ for a semimetric $d$, denoted by $N(\varepsilon, \Omega, d)$, is the minimal number of $d$-balls of radius $\varepsilon$ needed to cover the set $\Omega$. We denote by $\phi(\cdot)$ the standard normal density and by $P_f^n = \bigotimes P_{f,i}$ the $n$-fold product measure of the $n$ independent observations under (1) with a regression function $f(\cdot)$. By $\mathbb{P}_n^x = \frac{1}{n} \sum_{i=1}^n \delta_{x_i}$ we denote the empirical distribution of the observed covariates, by $||\cdot||_n$ the norm on $L_2(\mathbb{P}_n^x)$ and by $||\cdot||_2$ the standard Euclidean norm.

## 2  Bayesian Histogram Regression

We consider a classical nonparametric regression model, where response variables $\boldsymbol{Y}^{(n)} = (Y_1, \ldots, Y_n)'$ are related to input variables $\boldsymbol{x}^{(n)} = (x_1, \ldots, x_n)'$ through the function $f_0$ as follows

$$Y_i = f_0(x_i) + \varepsilon_i, \quad \varepsilon_i \sim \mathcal{N}(0, 1), \quad i = 1, \ldots, n. \tag{1}$$

We assume that the covariate values $x_i$ are one-dimensional, fixed and have been rescaled so that $x_i \in [0, 1]$. Partitioning-based regression methods are often invariant to monotone transformations of observations. In particular, when $f_0$ is a step function, standardizing the distance between the observations, and thereby the split points, has no effect on the nature of the estimation problem. Without loss of generality, we will thereby assume that the observations are aligned on an equispaced grid.

**Assumption 1.** *(Equispaced Grid) We assume that the scaled predictor values satisfy $x_i = \frac{i}{n}$ for each $i = 1, \ldots, n$.*

This assumption implies that partitions that are balanced in terms of the Lebesque measure will be balanced also in terms of the number of observations. A similar assumption was imposed by Donoho [28] in his study of Dyadic CART.

The underlying regression function $f_0 : [0, 1] \to \mathbb{R}$ is assumed to be a step function, i.e.

$$f_0(x) = \sum_{k=1}^{K_0} \beta_k^0 \mathbb{I}_{\Omega_k^0}(x),$$

where $\{\Omega_k^0\}_{k=1}^{K_0}$ is a partition of $[0, 1]$ into $K_0$ non-overlapping intervals. We assume that $\{\Omega_k^0\}_{k=1}^{K_0}$ is minimal, meaning that $f_0$ cannot be represented with a smaller partition (with less than $K_0$ pieces). Each partitioning cell $\Omega_k^0$ is associated with a step size $\beta_k^0$, determining the level of the function $f_0$ on $\Omega_k^0$. The entire vector of $K_0$ step sizes will be denoted by $\boldsymbol{\beta}^0 = (\beta_1^0, \ldots, \beta_K^0)'$.

One might like to think of $f_0$ as a regression tree with $K_0$ bottom leaves. Indeed, every step function can be associated with an equivalence class of trees that live on the same partition but differ in their tree topology. The number of bottom leaves $K_0$ will be treated as unknown throughout this paper. Our goal will be designing a suitable class of priors on step functions so that the posterior concentrates tightly around $f_0$. Our analysis with a single predictor has served as a precursor to a full-blown analysis for high-dimensional regression trees [29].

We consider an approximating space of all step functions (with $K = 1, 2, \ldots$ bottom leaves)

$$\mathcal{F} = \cup_{K=1}^{\infty} \mathcal{F}_K, \tag{2}$$

which consists of smaller spaces (or shells) of all $K$-step functions

$$\mathcal{F}_K = \left\{ f_{\boldsymbol{\beta}} : [0, 1] \to \mathbb{R}; f_{\boldsymbol{\beta}}(x) = \sum_{k=1}^{K} \beta_k \mathbb{I}_{\Omega_k}(x) \right\},$$

each indexed by a partition $\{\Omega_k\}_{k=1}^{K}$ and a vector of $K$ step heights $\boldsymbol{\beta}$. The fundamental building block of our theoretical analysis will be the prior on $\mathcal{F}$. This prior distribution has three main ingredients, described in detail below, (a) a prior on the number of steps $K$, (b) a prior on the partitions $\{\Omega_k\}_{k=1}^{K}$ of size $K$, and (c) a prior on step sizes $\boldsymbol{\beta} = (\beta_1, \ldots, \beta_K)'$.

## 2.1 Prior $\pi_K(\cdot)$ on the Number of Steps $K$

To avoid overfitting, we assign an exponentially decaying prior distribution that penalizes partitions with too many jumps.

**Definition 2.1.** *(Prior on $K$) The prior on the number of partitioning cells $K$ satisfies*

$$\pi_K(k) \equiv \Pi(K = k) \propto \exp(-c_K k \log k) \quad for \quad k = 1, 2, \ldots. \tag{3}$$

This prior is no stranger to non-parametric problems. It was deployed for stepwise reconstructions of densities [24, 23] and regression surfaces [25]. When $c_K$ is large, this prior is concentrated on models with small complexity where overfitting should not occur. Decreasing $c_K$ leads to the smearing of the prior mass over partitions with more jumps. This is illustrated in Figure 1, which depicts the prior for various choices of $c_K$. We provide recommendations for the choice of $c_K$ in Section 3.1.

## 2.2 Prior $\pi_\Omega(\cdot \mid K)$ on Interval Partitions $\{\Omega_k\}_{k=1}^{K}$

After selecting the number of steps $K$ from $\pi_K(k)$, we assign a prior over interval partitions $\pi_\Omega(\cdot \mid K)$. We will consider two important special cases.

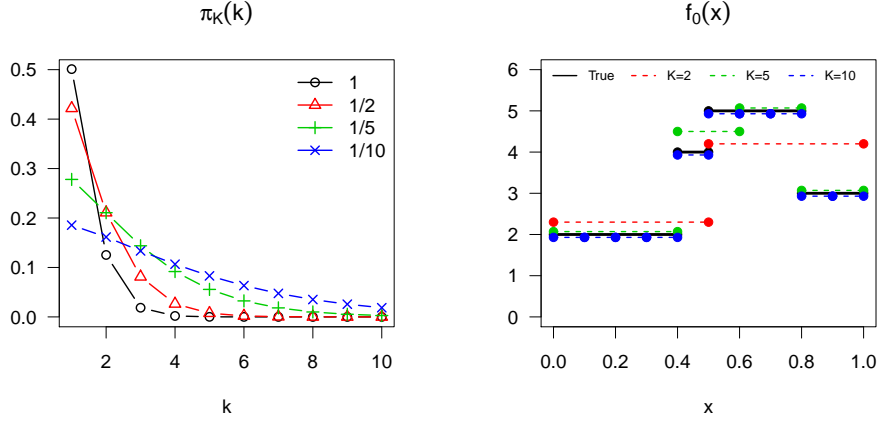

Figure 1: (Left) Prior on the tree size for several values of $c_K$, (Right) Best approximations of $f_0$ (in the $\ell_2$ sense) by step functions supported on equispaced blocks of size $K \in \{2, 5, 10\}$.

### 2.2.1 Equivalent Blocks

Perhaps the simplest partition is based on statistically equivalent blocks [30], where all the cells are required to have the same number of points. This is also known as the $K$-spacing rule that partitions the unit interval using order statistics of the observations.

**Definition 2.2.** *(Equivalent Blocks) Let $x_{(i)}$ denote the $i^{th}$ order statistic of $\boldsymbol{x} = (x_1, \dots, x_n)'$, where $x_{(n)} \equiv 1$ and $n = Kc$ for some $c \in \mathbb{N}\backslash\{0\}$. Denote by $x_{(0)} \equiv 0$. A partition $\{\Omega_k\}_{k=1}^K$ consists of $K$ equivalent blocks, when $\Omega_k = (x_{(j_k)}, x_{(j_{k+1})}]$, where $j_k = (k-1)c$.*

A variant of this definition can be obtained in terms of interval lengths rather than numbers of observations.

**Definition 2.3.** *(Equispaced Blocks) A partition $\{\Omega_k\}_{k=1}^K$ consists of $K$ equispaced blocks $\Omega_k$, when $\Omega_k = \left(\frac{k-1}{K}, \frac{k}{K}\right]$ for $k = 1, \dots, K$.*

When $K = 2^s$ for some $s \in \mathbb{N}\backslash\{0\}$, the equispaced partition corresponds to a full complete binary tree with splits at dyadic rationals. If the observations $x_i$ lie on a regular grid (Assumption 1), then Definition 2.2 and 2.3 are essentially equivalent. We will thereby focus on equivalent blocks (EB) and denote such a partition (for a given $K > 0$) with $\boldsymbol{\Omega}_K^{EB}$. Because there is only one such partition for each $K$, the prior $\pi_\Omega(\cdot|K)$ has a single point mass mass at $\boldsymbol{\Omega}_K^{EB}$. With $\boldsymbol{\Omega}^{EB} = \cup_{K=1}^\infty \boldsymbol{\Omega}_K^{EB}$ we denote the set of all EB partitions for $K = 1, 2, \dots$. We will use these partitioning schemes as a jump-off point.

### 2.2.2 Balanced Intervals

Equivalent (equispaced) blocks are deterministic and, as such, do not provide much room for learning about the actual location of jumps in $f_0$. Balanced intervals, introduced below, are a richer class of partitions that tolerate a bit more imbalance. First, we introduce the notion of cell counts $\mu(\Omega_k)$. For each interval $\Omega_k$, we write

$$\mu(\Omega_k) = \frac{1}{n}\sum_{i=1}^{n} \mathbb{I}(x_i \in \Omega_k), \tag{4}$$

the proportion of observations falling inside $\Omega_k$. Note that for equivalent blocks, we can write $\mu(\Omega_1) = \cdots = \mu(\Omega_K) = c/n = 1/K$.

**Definition 2.4.** *(Balanced Intervals) A partition $\{\Omega_k\}_{k=1}^K$ is balanced if*

$$\frac{C_{min}^2}{K} \le \mu(\Omega_k) \le \frac{C_{max}^2}{K} \quad for \ all \quad k = 1, \dots, K \tag{5}$$

*for some universal constants $C_{min} \le 1 \le C_{max}$ not depending on $K$.*

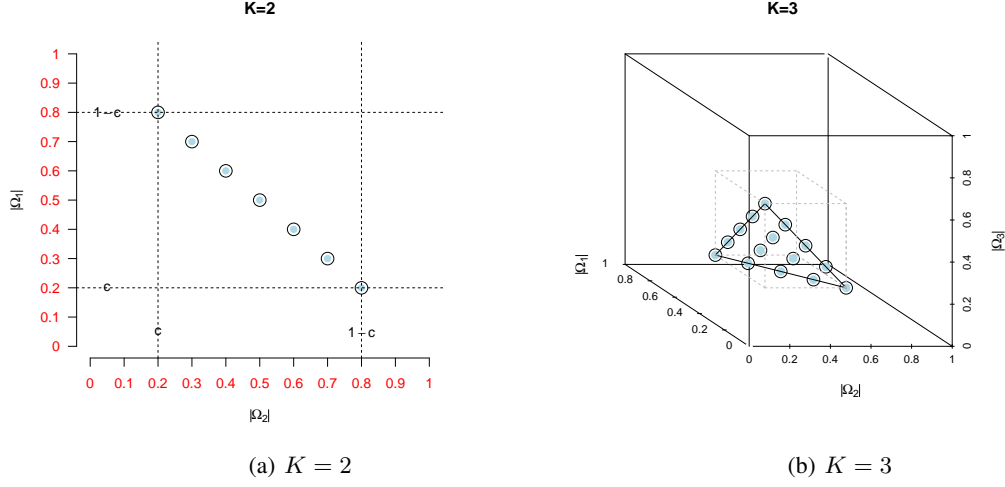

(a) $K = 2$  (b) $K = 3$

Figure 2: Two sets $E_K$ of possible stick lengths that satisfy the minimal cell-size condition $|\Omega_k| \geq C$ with $n = 10, C = 2/n$ and $K = 2, 3$.

The following variant of the balancing condition uses interval widths rather than cell counts: $\widetilde{C}_{min}^2/K \leq |\Omega_k| \leq \widetilde{C}_{max}^2/K$. Again, under Assumption 1, these two definitions are equivalent. In the sequel, we will denote by $\mathbf{\Omega}_K^{BI}$ the set of all balanced partitions consisting of $K$ intervals and by $\mathbf{\Omega}^{BI} = \cup_{K=1}^\infty \mathbf{\Omega}_K^{BI}$ the set of all balanced intervals of sizes $K = 1, 2, \ldots$. It is worth pointing out that the balance assumption on the interval partitions can be relaxed, at the expense of a log factor in the concentration rate [29].

With balanced partitions, the $K^{th}$ shell $\mathcal{F}_K$ of the approximating space $\mathcal{F}$ in (2) consists of all step functions that are supported on partitions $\mathbf{\Omega}_K^{BI}$ and have $K-1$ points of discontinuity $u_k \in I_n \equiv \{x_i : i = 1, \ldots, n-1\}$ for $k = 1, \ldots K - 1$. For equispaced blocks in Definition 2.3, we assumed that the points of subdivision were *deterministic*, i.e. $u_k = k/K$. For balanced partitions, we assume that $u_k$ are *random* and chosen amongst the *observed values* $x_i$. The order statistics of the vector of splits $\boldsymbol{u} = (u_1, \ldots, u_{K-1})'$ uniquely define a segmentation of $[0, 1]$ into $K$ intervals $\Omega_k = (u_{(k-1)}, u_{(k)}]$, where $u_{(k)}$ designates the $k^{th}$ smallest value in $\boldsymbol{u}$ and $u_{(0)} \equiv 0, u_{(K)} = x_{(n)} \equiv 1$.

Our prior over balanced intervals $\pi_\Omega(\cdot \mid K)$ will be defined implicitly through a *uniform* prior over the split vectors $\boldsymbol{u}$. Namely, the prior over balanced partitions $\mathbf{\Omega}_K^{BI}$ satisfies

$$\pi_\Omega(\{\Omega_k\}_{k=1}^K \mid K) = \frac{1}{\mathrm{card}(\mathbf{\Omega}_K^{BI})} \mathbb{I}\left(\{\Omega_k\}_{k=1}^K \in \mathbf{\Omega}_K^{BI}\right). \tag{6}$$

In the following Lemma, we obtain upper bounds on $\mathrm{card}(\mathbf{\Omega}_K^{BI})$ and discuss how they relate to an old problem in geometric probability. In the sequel, we denote with $|\Omega_k|$ the lengths of the segments defined through the split points $\boldsymbol{u}$.

**Lemma 2.1.** *Assume that $\boldsymbol{u} = (u_1, \ldots, u_{K-1})'$ is a vector of independent random variables obtained by uniform sampling (without replacement) from $I_n$. Then under Assumption 1, we have for $1/n < C < 1/K$*

$$\Pi\left(\min_{1 \leq k \leq K} |\Omega_k| \geq C\right) = \frac{\binom{\lfloor n(1-KC) \rfloor + K - 1}{K-1}}{\binom{n-1}{K-1}} \tag{7}$$

*and*

$$\Pi\left(\max_{1 \leq k \leq K} |\Omega_k| \leq C\right) = 1 - \sum_{k=1}^{\widetilde{n}}(-1)^k \binom{n-1}{k} \frac{\binom{\lfloor n(1-kC) \rfloor + K - 1}{K-1}}{\binom{n-1}{K-1}}, \tag{8}$$

*where $\widetilde{n} = \min\{n-1, \lfloor 1/C \rfloor\}$.*

*Proof.* The denominator of (7) follows from the fact that there are $n - 1$ possible splits for the $K - 1$ points of discontinuity $u_k$. The numerator is obtained after adapting the proof of Lemma

2 of Flatto and Konheim [31]. Without lost of generality, we will assume that $C = a/n$ for some $a = 1, \dots, \lfloor n/K \rfloor$ so that $n(1 - KC)$ is an integer. Because the jumps $u_k$ can only occur on the grid $I_n$, we have $|\Omega_k| = j/n$ for some $j = 1, \dots, n - 1$. It follows from Lemma 1 of Flatto and Konheim [31] that the set $E_K = \{|\Omega_k| : \sum_{k=1}^{K} |\Omega_k| = 1 \text{ and } |\Omega_k| \geq C \text{ for } k = 1, \dots, K\}$ lies in the interior of a convex hull of $K$ points $v_r = (1 - KC)e_r + C \sum_{k=1}^{K} e_k$ for $r = 1, \dots, K$, where $e_r = (e_{r1}, \dots, e_{rK})'$ are unit base vectors, i.e. $e_{rj} = \mathbb{I}(r = j)$. Two examples of the set $E_K$ (for $K = 2$ and $K = 3$) are depicted in Figure 2. In both figures, $n = 10$ (i.e. 9 candidate split points) and $a = 2$. With $K = 2$ (Figure 2(a)), there are only $7 = \binom{n(1-KC)+K-1}{K-1}$ pairs of interval lengths $(|\Omega_1|, |\Omega_2|)'$ that satisfy the minimal cell condition. These points lie on a grid between the two vertices $v_1 = (1 - C, C)$ and $v_2 = (C, 1 - C)$. With $K = 3$, the convex hull of points $v_1 = (1 - 2C, C, C)', v_2 = (C, 1 - 2C, C)'$ and $v_1 = (C, C, 1 - 2C)'$ corresponds to a diagonal dissection of a cube of a side length $(1 - 3C)$ (Figure 2(b), again with $a = 2$ and $n = 10$). The number of lattice points in the interior (and on the boundary) of such tetrahedron corresponds to an arithmetic sum $\frac{1}{2}(n - 3a + 2)(n - 3a + 1) = \binom{n-3a+2}{2}$. So far, we showed (7) for $K = 2$ and $K = 3$. To complete the induction argument, suppose that the formula holds for some arbitrary $K > 0$. Then the size of the lattice inside (and on the boundary) of a $(K + 1)$-tetrahedron of a side length $[1 - (K + 1)C]$ can be obtained by summing lattice sizes inside $K$-tetrahedrons of increasing side lengths $0, \sqrt{2}/n, 2\sqrt{2}/n, \dots, [1 - (K + 1)C]\sqrt{2}/n$, i.e.

$$\sum_{j=K-1}^{n[1-(K+1)C]+K-1} \binom{j}{K-1} = \binom{n[1 - (K + 1)C] + K}{K},$$

where we used the fact $\sum_{j=K}^{N} \binom{j}{K} = \binom{N+1}{K+1}$. The second statement (8) is obtained by writing the event as a complement of the union of events and applying the method of inclusion-exclusion. $\square$

**Remark 2.1.** *Flatto and Konheim [31] showed that the probability of covering a circle with random arcs of length $C$ is equal to the probability that all segments of the unit interval, obtained with iid random uniform splits, are smaller than $C$. Similarly, the probability* (8) *could be related to the probability of covering the circle with random arcs whose endpoints are chosen from a grid of $n - 1$ equidistant points on the circumference.*

There are $\binom{n-1}{K-1}$ partitions of size $K$, of which $\binom{\lfloor n(1-\widetilde{C}_{min}^2)\rfloor+K-1}{K-1}$ satisfy the minimal cell width balancing condition (where $\widetilde{C}_{min}^2 > K/n$). This number gives an upper bound on the combinatorial complexity of balanced partitions $\mathrm{card}(\boldsymbol{\Omega}_K^{BI})$.

### 2.3 Prior $\pi(\boldsymbol{\beta} \mid K)$ on Step Heights $\boldsymbol{\beta}$

To complete the prior on $\mathcal{F}^K$, we take independent normal priors on each of the coefficients. Namely

$$\pi(\boldsymbol{\beta} \mid K) = \prod_{k=1}^{K} \phi(\beta_k), \tag{9}$$

where $\phi(\cdot)$ is the standard normal density.

## 3 Main Results

A crucial ingredient of our proof will be understanding how well one can approximate $f_0$ with other step functions (supported on partitions $\boldsymbol{\Omega}$, which are either equivalent blocks $\boldsymbol{\Omega}^{EB}$ or balanced partitions $\boldsymbol{\Omega}^{BI}$). We will describe the approximation error in terms of the overlap between the true partition $\{\Omega_k^0\}_{k=1}^{K_0}$ and the approximating partitions $\{\Omega_k\}_{k=1}^{K} \in \boldsymbol{\Omega}$. More formally, we define the *restricted cell count* (according to Nobel [32]) as

$$m\left(V; \{\Omega_k^0\}_{k=1}^{K_0}\right) = |\Omega_k^0 : \Omega_k^0 \cap V \neq \emptyset|,$$

the number of cells in $\{\Omega_k^0\}_{k=1}^{K_0}$ that overlap with an interval $V \subset [0, 1]$. Next, we define the *complexity* of $f_0$ as the smallest size of a partition in $\boldsymbol{\Omega}$ needed to completely cover $f_0$ without any overlap.

**Definition 3.1.** *(Complexity of $f_0$ w.r.t. $\boldsymbol{\Omega}$) We define $K(f_0, \boldsymbol{\Omega})$ as the smallest $K$ such that there exists a $K$-partition $\{\Omega_k\}_{k=1}^K$ in the class of partitions $\boldsymbol{\Omega}$ for which*

$$m\left(\Omega_k; \{\Omega_k^0\}_{k=1}^{K_0}\right) = 1 \quad \text{for all} \quad k = 1, \dots, K.$$

*The number $K(f_0, \boldsymbol{\Omega})$ will be referred to as the complexity of $f_0$ w.r.t. $\boldsymbol{\Omega}$.*

The complexity number $K(f_0, \boldsymbol{\Omega})$ indicates the optimal number of steps needed to approximate $f_0$ with a step function (supported on partitions in $\boldsymbol{\Omega}$) without any error. It depends on the true number of jumps $K_0$ as well as the true interval lengths $|\Omega_k^0|$. If the minimal partition $\{\Omega_k^0\}_{k=1}^{K_0}$ resided in the approximating class, i.e. $\{\Omega_k^0\}_{k=1}^{K_0} \in \boldsymbol{\Omega}$, then we would obtain $K(f_0, \boldsymbol{\Omega}) = K_0$, the true number of steps. On the other hand, when $\{\Omega_k^0\}_{k=1}^{K_0} \notin \boldsymbol{\Omega}$, the complexity number $K(f_0, \boldsymbol{\Omega})$ can be much larger. This is illustrated in Figure 1 (right), where the true partition $\{\Omega_k^0\}_{k=1}^{K_0}$ consists of $K_0 = 4$ unequal pieces and we approximate it with equispaced blocks with $K = 2, 5, 10$ steps. Because the intervals $\Omega_k^0$ are not equal and the smallest one has a length $1/10$, we need $K(f_0, \boldsymbol{\Omega}^{EB}) = 10$ equispaced blocks to perfectly approximate $f_0$. For our analysis, we do not need to assume that $\{\Omega_k^0\}_{k=1}^{K_0} \in \boldsymbol{\Omega}$ (i.e. $f_0$ does not need to be inside the approximating class) or that $K(f_0, \boldsymbol{\Omega})$ is finite. The complexity number can increase with $n$, where sharper performance is obtained when $f_0$ can be approximated error-free with some $f \in \boldsymbol{\Omega}$, where $f$ has a small number of discontinuities relative to $n$.

Another way to view $K(f_0, \boldsymbol{\Omega})$ is as the ideal partition size on which the posterior should concentrate. If this number were known, we could achieve a near-minimax posterior concentration rate $n^{-1/2}\sqrt{K(f_0, \boldsymbol{\Omega}) \log[n/K(f_0, \boldsymbol{\Omega})]}$ (Remark 3.3). The actual minimax rate for estimating a piece-wise constant $f_0$ (consisting of $K_0 > 2$ pieces) is $n^{-1/2}\sqrt{K_0 \log(n/K_0)}$ [33]. In our main results, we will target the nearly optimal rate expressed in terms of $K(f_0, \boldsymbol{\Omega})$.

### 3.1 Posterior Concentration for Equivalent Blocks

Our first result shows that the minimax rate is nearly achieved, without any assumptions on the number of pieces of $f_0$ or the sizes of the pieces.

**Theorem 3.1.** *(Equivalent Blocks) Let $f_0 : [0, 1] \to \mathbb{R}$ be a step function with $K_0$ steps, where $K_0$ is unknown. Denote by $\mathcal{F}$ the set of all step functions supported on equivalent blocks, equipped with priors $\pi_K(\cdot)$ and $\pi(\boldsymbol{\beta} \mid K)$ as in (3) and (9). Denote with $K_{f_0} \equiv K(f_0, \boldsymbol{\Omega}^{EB})$ and assume $\|\boldsymbol{\beta}^0\|_\infty^2 \lesssim \log n$ and $K_{f_0} \lesssim \sqrt{n}$. Then, under Assumption 1, we have*

$$\Pi\left(f \in \mathcal{F} : \|f - f_0\|_n \geq M_n n^{-1/2}\sqrt{K_{f_0} \log(n/K_{f_0})} \mid \boldsymbol{Y}^{(n)}\right) \to 0 \tag{10}$$

*in $P_{f_0}^n$-probability, for every $M_n \to \infty$ as $n \to \infty$.*

Before we proceed with the proof, a few remarks ought to be made. First, it is worthwhile to emphasize that the statement in Theorem 3.1 is a frequentist one as it relates to an aggregated behavior of the posterior distributions obtained under the true generative model $P_{f_0}^n$.

Second, the theorem shows that the Bayesian procedure performs an automatic adaptation to $K(f_0, \boldsymbol{\Omega}^{EB})$. The posterior will concentrate on EB partitions that are fine enough to approximate $f_0$ well. Thus, we are able to recover the true function as well as if we knew $K(f_0, \boldsymbol{\Omega}^{EB})$.

Third, it is worth mentioning that, under Assumption 1, Theorem 3.1 holds for equivalent as well as equisized blocks. In this vein, it describes the speed of posterior concentration for *dyadic regression trees*. Indeed, as mentioned previously, with $K = 2^s$ for some $s \in \mathbb{N}\setminus\{0\}$, the equisized partition corresponds to a full binary tree with splits at dyadic rationals.

Another interesting insight is that the Gaussian prior (9), while selected for mathematical convenience, turns out to be sufficient for optimal recovery. In other words, despite the relatively large amount of mass near zero, the Gaussian prior does not rule out optimal posterior concentration. Our standard normal prior is a simpler version of the Bayesian CART prior, which determines the variance from the data [9].

Let $K_{f_0} \equiv K(f_0, \boldsymbol{\Omega}^{EB})$ be as in Definition 3.1. Theorem 3.1 is proved by verifying the three conditions of Theorem 4 of [18], for $\varepsilon_n = n^{-1/2}\sqrt{K_{f_0} \log(n/K_{f_0})}$ and $\mathcal{F}_n = \bigcup_{K=0}^{k_n} \mathcal{F}_K$, with

$k_n$ of the order $K_{f_0}\log(n/K_{f_0})$. The approximating subspace $\mathcal{F}_n \subset \mathcal{F}$ should be rich enough to approximate $f_0$ well and it should receive most of the prior mass. The conditions for posterior contraction at the rate $\varepsilon_n$ are:

(C1) $\displaystyle\sup_{\varepsilon > \varepsilon_n} \log N\left(\frac{\varepsilon}{36}, \{f \in \mathcal{F}_n : \|f - f_0\|_n < \varepsilon\}, \|.\|_n\right) \leq n\varepsilon_n^2,$

(C2) $\displaystyle\frac{\Pi(\mathcal{F}\backslash\mathcal{F}_n)}{\Pi(f \in \mathcal{F} : \|f - f_0\|_n^2 \leq \varepsilon_n^2)} = o\left(e^{-2n\varepsilon_n^2}\right),$

(C3) $\displaystyle\frac{\Pi(f \in \mathcal{F}_n : j\varepsilon_n < \|f - f_0\|_n \leq 2j\varepsilon_n)}{\Pi(f \in \mathcal{F} : \|f - f_0\|_n^2 \leq \varepsilon_n^2)} \leq e^{\frac{j^2}{4}n\varepsilon_n^2}$ for all sufficiently large $j$.

The entropy condition (C1) restricts attention to EB partitions with small $K$. As will be seen from the proof, the largest allowed partitions have at most (a constant multiple of) $K_{f_0}\log(n/K_{f_0})$ pieces..

Condition (C2) requires that the prior does not promote partitions with more than $K_{f_0}\log(n/K_{f_0})$ pieces. This property is guaranteed by the exponentially decaying prior $\pi_K(\cdot)$, which penalizes large partitions.

The final condition, (C3), requires that the prior charges a $\|.\|_n$ neighborhood of the true function. In our proof, we verify this condition by showing that the prior mass on step functions of the optimal size $K_{f_0}$ is sufficiently large.

*Proof.* We verify the three conditions (C1), (C2) and (C3).

**(C1)** Let $\varepsilon > \varepsilon_n$ and $K \in \mathbb{N}$. For $f_{\boldsymbol\alpha}, f_{\boldsymbol\beta} \in \mathcal{F}_K$, we have $K^{-1}\|\boldsymbol\alpha - \boldsymbol\beta\|_2^2 = \|f_{\boldsymbol\alpha} - f_{\boldsymbol\beta}\|_n^2$ because $\mu(\Omega_k) = 1/K$ for each $k$. We now argue as in the proof of Theorem 12 of [18] to show that $N\left(\frac{\varepsilon}{36}, \{f \in \mathcal{F}_K : \|f - f_0\|_n < \varepsilon\}, \|.\|_n\right)$ can be covered by the number of $\sqrt{K}\varepsilon/36$-balls required to cover a $\sqrt{K}\varepsilon$-ball in $\mathbb{R}^K$. This number is bounded above by $108^K$. Summing over $K$, we recognize a geometric series. Taking the logarithm of the result, we find that (C1) is satisfied if $\log(108)(k_n + 1) \leq n\varepsilon_n^2$.

**(C2)** We bound the denominator by:

$$\Pi(f \in \mathcal{F} : \|f - f_0\|_n^2 \leq \varepsilon^2) \geq \pi_K(K_{f_0})\Pi\left(\boldsymbol\beta \in \mathbb{R}^{K(f_0)} : \|\boldsymbol\beta - \boldsymbol\beta_0^{\text{ext}}\|_2^2 \leq \varepsilon^2 K_{f_0}\right),$$

where $\boldsymbol\beta_0^{\text{ext}} \in \mathbb{R}^{K_{f_0}}$ is an extended version of $\boldsymbol\beta^0 \in \mathbb{R}^{K_0}$, containing the coefficients for $f_0$ expressed as a step function on the partition $\{\Omega_k^0\}_{k=1}^{K_{f_0}}$. This can be bounded from below by

$$\frac{\pi_K(K_{f_0})}{e^{\|\boldsymbol\beta_0^{\text{ext}}\|_2^2/2}}\Pi\left(\boldsymbol\beta \in \mathbb{R}^{K(f_0)} : \|\boldsymbol\beta\|_2^2 \leq \varepsilon^2 K_{f_0}/2\right) > \frac{\pi_K(K_{f_0})}{e^{\|\boldsymbol\beta_0^{\text{ext}}\|_2^2/2}}\int_0^{\varepsilon^2 K_{f_0}/2}\frac{x^{K_{f_0}/2-1}e^{-x/2}}{2^{K_{f_0}/2}\Gamma(K_{f_0}/2)}dx.$$

We bound this from below by bounding the exponential at the upper integration limit, yielding:

$$\frac{\pi_K(K_{f_0})}{e^{\|\boldsymbol\beta_0^{\text{ext}}\|_2^2/2}}\frac{e^{-\varepsilon^2 K_{f_0}/4}}{2^{K_{f_0}}\Gamma(K_{f_0}/2 + 1)}\varepsilon^{K_{f_0}}K_{f_0}^{K_{f_0}/2}. \tag{11}$$

For $\varepsilon = \varepsilon_n \to 0$, we thus find that the denominator in (C2) can be lower bounded with $e^{K_{f_0}\log\varepsilon_n - c_K K_{f_0}\log K_{f_0} - \|\boldsymbol\beta_0^{ext}\|_2^2/2 - K_{f_0}/2[\log 2 + \varepsilon_n^2/2]}$. We bound the numerator:

$$\Pi(\mathcal{F}\backslash\mathcal{F}_n) = \Pi\left(\bigcup_{k=k_n+1}^{\infty}\mathcal{F}_k\right) \propto \sum_{k=k_n+1}^{\infty}e^{-c_K k\log k} \leq e^{-c_K(k_n+1)\log(k_n+1)} + \int_{k_n+1}^{\infty}e^{-c_K x\log x},$$

which is of order $e^{-c_K(k_n+1)\log(k_n+1)}$. Combining this bound with (11), we find that (C2) is met if:

$$e^{-K_{f_0}\log\varepsilon_n + (c_K+1)K_{f_0}\log K_{f_0} + K_{f_0}\|\boldsymbol\beta^0\|_\infty^2 - c_K(k_n+1)\log(k_n+1) + 2n\varepsilon_n^2} \to 0 \text{ as } n \to \infty.$$

**(C3)** We bound the numerator by one, and use the bound (11) for the denominator. As $\varepsilon_n \to 0$, we obtain the condition $-K_{f_0}\log\varepsilon_n + (c_K+1)K_{f_0}\log K_{f_0} + K_{f_0}\|\boldsymbol\beta^0\|_\infty^2 \leq \frac{j^2}{4}n\varepsilon_n^2$ for all sufficiently large $j$.

**Conclusion** With $\varepsilon_n = n^{-1/2}\sqrt{K_{f_0}\log(n/K_{f_0})}$, letting $k_n \propto n\varepsilon_n^2 = K_{f_0}\log(n/K_{f_0})$, the condition (C1) is met. With this choice of $k_n$, the condition (C2) holds as well as long as $\|\boldsymbol{\beta}^0\|_\infty^2 \lesssim \log n$ and $K_{f_0} \lesssim \sqrt{n}$. Finally, the condition (C3) is met for $K_{f_0} \lesssim \sqrt{n}$.  $\square$

**Remark 3.1.** *It is worth pointing out that the proof will hold for a larger class of priors on $K$, as long as the prior shrinks at least exponentially fast (meaning that it is bounded from above by $ae^{-bK}$ for constants $a, b > 0$). However, a prior at this exponential limit will require tuning, because the optimal $a$ and $b$ will depend on $K(f_0, \boldsymbol{\Omega}^{EB})$. We recommend using the prior* (2.1) *that prunes somewhat more aggressively, because it does not require tuning by the user. Indeed, Theorem 3.1 holds regardless of the choice of $c_K > 0$. We conjecture, however, that values $c_K \geq 1/K(f_0, \boldsymbol{\Omega}^{EB})$ lead to a faster concentration speed and we suggest $c_K = 1$ as a default option.*

**Remark 3.2.** *When $K_{f_0}$ is known, there is no need for assigning a prior $\pi_K(\cdot)$ and the conditions (C1) and (C3) are verified similarly as before, fixing the number of steps at $K_{f_0}$.*

## 3.2 Posterior Concentration for Balanced Intervals

An analogue of Theorem 3.1 can be obtained for balanced partitions from Section 2.2.2 that correspond to regression trees with splits at actual observations. Now, we assume that $f_0$ is $\boldsymbol{\Omega}^{BI}$-valid and carry out the proof with $K(f_0, \boldsymbol{\Omega}^{BI})$ instead of $K(f_0, \boldsymbol{\Omega}^{EB})$. The posterior concentration rate is only slightly worse.

**Theorem 3.2.** *(Balanced Intervals) Let $f_0 : [0, 1] \to \mathbb{R}$ be a step function with $K_0$ steps, where $K_0$ is unknown. Denote by $\mathcal{F}$ the set of all step functions supported on balanced intervals equipped with priors $\pi_K(\cdot), \pi_\Omega(\cdot|K)$ and $\pi(\boldsymbol{\beta} \mid K)$ as in* (3), (6) *and* (9). *Denote with $K_{f_0} \equiv K(f_0, \boldsymbol{\Omega}^{BI})$ and assume $\|\boldsymbol{\beta}^0\|_\infty^2 \lesssim \log^{2\beta} n$ and $K(f_0, \boldsymbol{\Omega}^{BI}) \lesssim \sqrt{n}$. Then, under Assumption 1, we have*

$$\Pi\left(f \in \mathcal{F} : \|f - f_0\|_n \geq M_n n^{-1/2}\sqrt{K_{f_0}\log^{2\beta}(n/K_{f_0})} \mid \boldsymbol{Y}^{(n)}\right) \to 0 \qquad (12)$$

*in $P_{f_0}^n$-probability, for every $M_n \to \infty$ as $n \to \infty$, where $\beta > 1/2$.*

*Proof.* All three conditions (C1), (C2) and (C3) hold if we choose $k_n \propto K_{f_0}[\log(n/K_{f_0})]^{2\beta-1}$. The entropy condition will be satisfied when $\log\left(\sum_{k=1}^{k_n} C^k \mathrm{card}(\boldsymbol{\Omega}_k^{BI})\right) \lesssim n\varepsilon_n^2$ for some $C > 0$, where $\varepsilon_n = n^{-1/2}\sqrt{K_{f_0}\log^{2\beta}(n/K_{f_0})}$. Using the upper bound $\mathrm{card}(\boldsymbol{\Omega}_k^{BI}) < \binom{n-1}{k-1} < \binom{n-1}{k_n-1}$ (because $k_n < \frac{n-1}{2}$ for large enough $n$), the condition (C1) is verified. Using the fact that $\mathrm{card}(\boldsymbol{\Omega}_{K_{f_0}}) \lesssim K_{f_0}\log(n/K_{f_0})$, the condition (C2) will be satisfied when, for some $D > 0$, we have

$$e^{-K_{f_0}\log\varepsilon_n + (c_K+1)K_{f_0}\log K_{f_0} + D K_{f_0}\log(n/K_{f_0}) + K_{f_0}\|\boldsymbol{\beta}^0\|_\infty^2 - c_K(k_n+1)\log(k_n+1) + 2n\varepsilon_n^2} \to 0. \quad (13)$$

This holds for our choice of $k_n$ under the assumption $\|\boldsymbol{\beta}^0\|_\infty^2 \lesssim \log^{2\beta} n$ and $K_{f_0} \lesssim \sqrt{n}$. These choices also yield (C3).  $\square$

**Remark 3.3.** *When $K_{f_0} \gtrsim \sqrt{n}$, Theorem 3.1 and Theorem 3.2 still hold, only with the bit slower slower concentration rate $n^{-1/2}\sqrt{K_{f_0}\log n}$.*

## 4 Discussion

We provided the first posterior concentration rate results for Bayesian non-parametric regression with step functions. We showed that under suitable complexity priors, the Bayesian procedure adapts to the unknown aspects of the target step function. Our approach can be extended in three ways: (a) to smooth $f_0$ functions, (b) to dimension reduction with high-dimensional predictors, (c) to more general partitioning schemes that correspond to methods like Bayesian CART and BART. These three extensions are developed in our followup manuscript [29].

## 5 Acknowledgment

This work was supported by the James S. Kemper Foundation Faculty Research Fund at the University of Chicago Booth School of Business.

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
