[Reviews · NeurIPS 2017]

Reviewer 1



The paper gives a detailed theory of a very simple case of regression trees, where there is one predictor variable. The main contribution is probably the methods used. There is a follow-up manuscript, and I'd say the more extensive results should have made the paper to make it more interesting.

Reviewer 2



This paper analyses concentration rates (speed of posterior concentration) for Bayesian regression histograms and demonstrates that under certain conditions and priors, the posterior distribution concentrates around the true step regression function at the minimax rate. The notation is clear. Different approximating functions are considered, starting from the set of step functions supported on equally sized intervals, up to more flexible functions supported on balanced partitions. The most important part of the paper is building the prior on the space of approximating functions. The paper is relatively clear and brings up an interesting first theoretical result regarding speed of posterior concentration for Bayesian regression histograms. The authors assume very simple conditions (one predictor, piecewise-constant functions), but this is necessary in order to get a first analysis. Proof seems correct, although it is out of this reviewer's expertise. This reviewer wonders why the authors did not considered sending this work to the Annals of Statistics instead of NIPS, given the type of analysis and provided results. Minor: - You might want to check the reference of Bayesian Hierarchical Clustering (Heller et.al, 2005) and the literature of infinite mixture of experts. - l. 116: You mention recommendations for the choice of c_K in Section 3.3, but this Section does not exist. - Assumption 1 is referred to multiple times in page 4, but it is not defined until page 5. Writing (typos): - l. 83: we will - l. 231: mass near zero - l. 296: (b) not clear (extend to multi-dimensional?)

Reviewer 3



In this paper, Authors focus on Bayesian regression histograms aiming at regression with one explanatory variable. They develop an approach for constructing regression surface that is piece-wise constant in which no.of jumps are unknown. The approach has some merits and I have some concerns given below, which I think authors should address for better clarity. 1. Line 116: There is no section 3.3 in the paper. 2. In the proof of lemma 2.1 in the supplementary file, it is not clear ( at least to me) how you arrived at the expression above line 24. I followed your explanation but couldn't get it. 3. In line 178: There should be a better explanation for assuming x's are fixed. What if not? 4. Line 183: Do you refer this xi to be the scaled predictor? Line: 296: Minor comment: How flexible to extend this to dimension reduction with many predictors? In a practical application, How one can select Ck? and How sensitive the theorem 3.1 to this choice?